# Broadband Plasmonic Polarization Filter Based on Photonic Crystal Fiber with Dual-Ring Gold Layer

**DOI:** 10.3390/mi11050470

**Published:** 2020-04-29

**Authors:** Nan Chen, Xuedian Zhang, Min Chang, Xinglian Lu, Jun Zhou

**Affiliations:** 1Key Laboratory of Optical Technology and Instrument for Medicine, Ministry of Education, University of Shanghai for Science and Technology, Shanghai 200093, China; 181560053@st.usst.edu.cn (N.C.); changmin@usst.edu.cn (M.C.); 151360021@st.usst.edu.cn (X.L.); 151360026@st.usst.edu.cn (J.Z.); 2Shanghai Key Laboratory of Molecular Imaging, Shanghai University of Medicine and Health Sciences, Shanghai 201318, China

**Keywords:** photonic crystal fiber, polarization filter, surface plasmon resonance, confinement loss, extinction ratio

## Abstract

Polarization filter is a very important optical device with extinction characteristics. Due to the design flexibility of photonic crystal fibers and the high excitation losses of the gold layer, the polarization filter based on the photonic crystal fiber and surface plasmonic resonance effect is widely studied. Considering these, we present a simple and high-performance polarization filter using the finite element method. Numerical simulations show that there is a large difference in energy between the two polarization directions by reasonable adjustment of the structural parameters, the confinement loss in the x-pol direction is less than that in the y-pol direction, which is suitable to realize a broadband polarization filter. When the fiber length is 2 mm, the extinction ratio peak can reach −478 dB, and the bandwidth with the extinction ratio better than −20 dB is 750 nm, which covers communication wavelengths of 1.31 μm and 1.55 μm (1.05–1.8 μm). It also has a low insertion loss of 0.11 dB at 1.31 μm and 0.04 dB at 1.55 μm. In addition, our design has high feasibility in fabrication and better tolerance. The proposed filter with compactness, high extinction ratio, broad bandwidth, and low insertion loss would play an important role in the sensing detection, bio-medical, and telecommunication field.

## 1. Introduction

Since the discovery of surface plasmon polariton [1] (SPP), SPP-based research has been implemented rapidly around the world as an exciting field of combining surface plasmon linking and nano-photonics research. As the name implies, SPP refers to electromagnetic waves that propagate along the surface of the metal-dielectric interface in the infrared or visible light band [2,3,4,5]. SPP has excellent electromagnetic energy binding properties and can break the diffraction limit, so it has become a new round of research hotspot. Due to the presence of SPP, the surface plasmon resonance (SPR) effect appears in the metal-dielectric interface. SPR can generate a large amount of energy loss that can be widely used in many fields, such as near-field optics, bio-sensing, superconducting materials, etc. Various plasmonic optical devices can be fabricated with this feature.

In recent years, photonic crystal fiber [6,7] (PCF), as a hot research field, is widely employed in new fiber optic equipment because of its flexible design, endless single-mode transmission, wide tuning range, good temperature stability, as well as micro-nano level dimension. The PCF-based polarization filter is one of the most popular optical applications [8]. Polarization filters are optical devices that separate an incident light by its polarization characteristics, and are mainly used in optical systems that require the effective control of the polarization state [9,10]. Especially in conjunction with SPR technology, the PCF filter research becomes richer. To date, many experts and scholars have proposed many novel PCF-SPR filter designs. For example, in 2015, Hameed et al. proposed a novel ultra-high tunable PCF polarization filter, the reported filter of compact device length 0.5 mm could achieve 600 dB/cm resonance losses at φ = 90° for the x-polarized (x-pol) mode at 1300 nm with low losses of 0.00751 dB/cm for the y-pol mode [11]. In the same year, they proposed and analyzed a selectively metal-filled spiral PCF with an elliptic core, at wavelength λ = 1013 nm, the loss of the x-pol core mode was equal to 77.04 dB/mm, while it is only 2.765 dB/mm for the y-pol core mode with a single metal rod [12]. Liu et al. proposed a broadband single-polarization PCF-SPR filter, and the fiber length was 3 mm, the bandwidth of extinction ratio better than −20 dB was greater than 430 nm covering almost all the communication wavelength [13]. In 2016, Chen et al. proposed a polarization filter based on liquid crystal infiltrated PCFs with a gold wire, the confinement loss could achieve 446 dB/cm in the y-pol and 0.8 dB/cm in the x-pol at 1550 nm [14]. An et al. designed a symmetry-structured PCF with gold-filled holes, at 1310 nm, the x-pol fundamental mode (FM) could be filtered with the metal wire *d_m_* = 1.2 μm, when *d_m_* = 1.4 μm, the y-pol FM could be filtered at 1.55 μm [15]. In 2017, Zhao et al. designed a new type of gold-coated PCF filter, at 1550 nm, the loss in the y-pol direction is 906.9 dB/cm, which is much larger than the x-pol direction. The fiber length was longer than 100 μm, the crosstalk was greater than 20 dB (1.4–1.9 μm) [16]. In 2018, Dou et al. proposed a PCF filter with the gold layer, which had a short length, high extinction ratio, and low insertion loss [17]. In 2019, Almewafy et al. proposed a SPR-PCF polarization filter with a compact device length of 23 μm, which can achieve good crosstalk (CT) of −46.4 and −41.2 dB at the communication wavelengths of 1.3 and 1.52 μm, respectively [18].

The PCF filters integrated with SPR possess better performance than the conventional filters that can be used in sensing, biomedicine, communications, etc., and the demand is gradually increasing, showing a high research value. From the above PCF-SPR filters, gold (Au) is the most commonly used excitation materials to produce the SPR phenomenon. Compared with other metals such as silver (Ag), copper (Cu), and aluminum (Al), gold can excite significant energy and has better stability, so filling the gold wire or coating the gold film in the PCF can achieve excellent filter performance of compactness, high extinction ratio, broad bandwidth, and low insertion loss. Therefore, our plasmonic PCF filter with the dual-ring gold layer is proposed, and the performance changes affected by the structural parameters are investigated by the finite element method (FEM) [19] and coupled-mode theory (CMT) [20]. What’s more, our design has high fabrication feasibility and better tolerance.

## 2. PCF Filter Model and Theory

Figure 1 shows the PCF filter cross-section, the proposed PCF preform structure is based on the stacking of different capillaries and then draw PCF to the desired dimension. The cladding is in the form of a triangular lattice, and the pitch *Λ* in the fiber is 2 μm. A total of three sizes of air holes are included, which are *d*_1_, *d*_2_, and *d*_3_ as indicated in the figure, where *d*_1_ = 1.2 μm. The ring of the holes of *t* are coated with the gold film.

The refractive index (RI) of air is set to 1, and the substrate material is fused silica, its material dispersion is already being considered using the Sellmeier equation [21,22,23].
(1)$$n_{fused\;silica}^{2}\left(\lambda \right)=1+B_{1}\lambda^{2}{\left(\lambda^{2}-C_{1}\right)}^{-1}+B_{2}\lambda^{2}{\left(\lambda^{2}-C_{2}\right)}^{-1}+B_{3}\lambda^{2}{\left(\lambda^{2}-C_{3}\right)}^{-1}$$
where *B*_1_ = 0.696163, *B*_2_ = 0.4079426, *B*_3_ = 0.8974794; *C*_1_ = 0.0046791 ${\mu m}^{2}$, *C*_2_ = 0.0135121 ${\mu m}^{2}$, *C*_3_ = 97.93400 ${\mu m}^{2}$, the RI of fused silica is between 1.4504 and 1.4408 in the investigated wavelength range and *λ* is the operating wavelength in vacuum. The material dispersion of gold is also characterized by a Drude–Lorentz (DL) model [24,25,26], which can be expressed as:(2)$$\varepsilon_{m}=\varepsilon_{\infty }-\frac{\omega_{D}^{2}}{\omega \left(\omega +j\gamma_{D}\right)}-\frac{\Delta \varepsilon \cdot \Omega_{L}^{2}}{\left(\omega^{2}-\Omega_{L}^{2}\right)-j{Г}_{L}\omega }$$
where $\varepsilon_{\infty }$ is the permittivity of gold, Δε can be interpreted as a weighting factor, and *ω* is the angular frequency of the guided light; $\omega_{D}$, $\gamma_{D}$, $\Omega_{L}$, and ${Г}_{L}$ represent the plasma frequency, damping frequency, the frequency, and the spectral width of the Lorentz oscillator, respectively. The detailed parameters in the DL model are shown in Table 1.

For SPR, when the energy and the momentum of the incident light and surface plasmon wavelength (SPW) match, in other words, the phase-matching condition (PMC) is satisfied, the resonance phenomenon occurs. The necessary condition for SPR [27,28] is given by:(3)$$\frac{2\pi }{\lambda }\sqrt{\varepsilon_{p}}sin\theta =\frac{2\pi }{\lambda }{\left(\frac{\varepsilon_{m}\varepsilon_{s}}{\varepsilon_{m}+\varepsilon_{s}}\right)}^{1/2}$$
where *λ* is the light wavelength, *θ* is the angle of the incident light, $\varepsilon_{p}$ represents the dielectric constant of the material on which metal is deposited, and $\varepsilon_{m}$ and $\varepsilon_{s}$ are the dielectric constants of the metal and the sensing medium, respectively. During the SPR effect, the excited energy will be transferred from the incident light to surface plasmons (SPs), as the effect of the evanescent field stationary wave becomes more significant; a sharp dip appears at the particular wavelength. In the paper, the transmission spectra of the proposed filter can be represented by confinement loss (*L_c_*) [29,30] which is expressed as:(4)$$L_{c}\left(dB/mm\right)=8.686\times \frac{2\pi }{\lambda }\times Im\left(n_{eff}\right)\times 10^{3}$$
where the unit of the *L_c_* is decibel per micro-meter, and $Im\left(n_{eff}\right)$ is the imaginary part of the effective RI $n_{eff}$. The FEM is used to analyze the characteristics of the PCF polarization filter and the perfectly matched layer [28] (PML) is employed to absorb radiant energy for improving the calculation precision, here, the thickness of the PML is 5 μm. The outer boundary of the PML is set to the scattering boundary condition to further reduce the reflecting energy. In the mature calculation Comsol-5.4 software integrated finite element solver, an electromagnetic hybrid wave perpendicular to the PCF end is adopted [31]. Additionally, there are 160 vertex elements, 1355 boundary elements and 16,584 mesh elements in the computational region.

## 3. Results and Analysis

### 3.1. Dispersion Relations

According to the literature [32,33], the SPR produced by the 2-SPP mode and the core FM is exactly located in the common communication windows and can produce a very strong resonance coupling effect, which is very suitable for polarization-dependent characteristic applications. In the PCF, the electric field distributions of SPP mode and FMs can be obtained from Figure 2. Through the numerical calculation, the dispersion relationship between the core FM and 2-SPP mode is obtained as depicted in Figure 3. It can be found that both the x- and y-pol state can resonant with different 2-SPP modes when the FM and 2-SPP mode are coupled, and the energy is exchanged to produce a very large dramatic change; as a result, two blue curve spikes emerge. We can also find that the coupling energy generated in the y-pol direction is significantly greater than that in the x-pol direction, which opens up possibilities for the proposed filter, so the design of the PCF-filter is performed.

### 3.2. Structural Parameter Effects

As we all know, the performance of PCFs is very sensitive to changes in structural parameters [34], and the transmission characteristics will change accordingly when structural parameters of the PCF change, so the choice of structural parameters directly determines the performance of this optical device. In this section, the effect of the pore diameter *d*_2_, *d*_3_, and the gold film thickness *t* are investigated numerically on the PCF filter.

Firstly, we discuss the impact of *d*_2_. As shown in Figure 4, when other parameters remain constant (*d*_1_ = 1.2 μm, *d*_3_ is 1.2 μm, and *t* is 100 nm), we make *d*_2_ 2 μm, 2.2 μm, and 2.4 μm, respectively. We can find that as the d_2_ increases, the corresponding x- and y-pol loss curves move toward the long-wavelength direction, the resonance peak value increases significantly, and the x- and y-pol resonance peak spacing increases under the same *d*_2_ condition. The *d*_2_ is larger than *d*_1_ the birefringence of this PCF is increased, but it does not mean that the larger *d*_2_, the better the performance. The results show that when *d*_2_ is 2.4μm, the curve in the y-pol direction no longer covers the x-pol direction, which is not conducive to generating broadband. Therefore, the intermediate value *d*_2_ = 2.2 μm is selected.

Secondly, we analyze the effect of gold film thickness *t* on the performance of the PCF-filter. The *t* values were taken at the nano-scale, 60 nm, 80 nm, and 100 nm, respectively. When *d*_1_ = 1.2 μm, *d*_2_ is 2.4 μm, and *d*_3_ is 1.2 μm. As shown in Figure 5, as the film thickness increases, the loss curve is blue-shifted, unlike the previous phenomenon, the energy in the y-pol direction increases. However, the energy in the x-pol direction is decreasing. In addition, the loss of peak spacing in both polarization directions is almost constant. We can fine-tune the performance of this filter based on the film thickness. When *t* is 100 nm, there is a larger difference in the peaks between the x- and y-pol direction, so we choose *t* = 100 nm for the next discussion.

Thirdly, we take into account the effect of different *d*_3_ and make *d*_3_ 0.9 μm, 1.2 μm, and 1.5 μm, respectively when *d*_1_ = 1.2 μm, *d*_2_ is 2.2 μm, and *t* is 100 nm. Figure 6 shows that because of the increase in d_3_, the loss peak is significantly reduced and the x- and y-pol peak spacing are slightly reduced. In addition, although the loss curves corresponding to different *d*_3_ values are significantly reduced, the difference between the x- and y-pol peaks is rather prominent. In the inset, it can be seen that when *d*_3_ = 1.5 μm, the loss peak in the y-pol direction is 23.4 dB/mm and the peak loss in the x-pol direction is 1.2 dB/mm. Although the energy level is not comparable to the other two cases, the energy difference in the two polarization directions is more than 10 times and the loss in the x-pol direction is less than that in the y-pol direction in the investigated wavelength range, which is suitable to produce effective polarization filter effect.

### 3.3. Filter Performance

According to the discussion in 3.2, when *d*_1_ = 1.2 μm, *d*_2_ = 2.2 μm, *d*_3_ = 1.5 μm, and *t* = 100 nm, the resonance peak in the y-pol direction is 23.4 dB/mm and the peak loss in the x-pol direction is 1.2 dB/mm, the energy difference in the two polarization directions is more than 10 times and the confinement loss curve of y-pol mode is larger than the loss of x-pol mode in the investigated wavelength range which makes the fiber possible to realize a broadband polarization filter. Normalized output power (NOP) [35], extinction ratio (ER) [36,37] and insertion loss (IL) [38] are employed for evaluating the performance of this PCF-filter. The NOP $P_{out}$ can be obtained by the equation:(5)$$P_{out}\left(x,y\right)=P_{in}\left(x,y\right)\times exp\left(-\alpha \left(x,y\right)\times L\times \frac{ln10}{10}\right)$$
where the input power *P_in_* is assumed to be 1, *L* represents the fiber length.

ER and IL are two key elements for a polarization filter which are calculated by the following equation:(6)$$ER=10log_{10}\left(\frac{P_{out}\left(y\right)}{P_{out}\left(x\right)}\right)$$
(7)$$IL_{y}=-10log_{10}\left(\frac{P_{out}\left(y\right)}{P_{in}}\right)$$
where *P_out_(x)* and *P_out_(y)* are x- and y-pol output power. The calculated results for these three performance parameters are shown in Figure 7 and Figure 8.

For NOP, all curves are distributed between 0 and 1, and the output power curve in the y-pol direction has two peaks. When the fiber length increases, the corresponding output power decreases overall. In the x-pol direction, the curve drops steeply and then flattens out, and the flatness increases as the length of the fiber increases. The corresponding ER curves are shown in Figure 7b; we have found that when the fiber length *L* increases, the ER increases and the extinction degree becomes higher. When *L* is 3 mm, ER can reach −698 dB. For IL, as the increment of *L*, the IL value increases. However, IL needs to be reduced as much as possible in this device, so we choose *L* = 2 mm as a compromise. When the fiber length is 2 mm, the ER peak can reach −478 dB. If the ER can be controlled below −20 dB, it means that the x- and y-pol modes can be divided well. According to this criterion, a surprising result can be found that the filter has a 750 nm bandwidth which covers the two communication windows of 1.31 and 1.55 μm (1.05–1.8μm). Besides, IL is 0.11 dB at 1.31 μm and 0.04 dB at 1.55 μm. Therefore, a broadband PCF polarization filter with compactness, high ER, broad bandwidth, and low IL can be achieved. In addition, the single-ring structure filter is also investigated as shown in Figure 8. The principles of two filters are similar, according to the mode field shown in Figure 8b, the single-ring structure will have a dispersion relationship similar to that in Figure 3, then the three performance parameters can be achieved. When *L* is 2 mm, the maximum ER is −256 dB, the bandwidth is about 530 nm (1.12–1.65 μm), and IL is below 0.1 dB at 1.31 μm and 1.55 μm. Overall, the proposed filter with dual-ring gold layer has a better performance.

For comparison, the prior filters and the proposed filter are listed in Table 2. The comparison results show that when the fiber length is 2 mm, the proposed filter has a high ER superior to others, very low IL, and broader bandwidth than the mostly devices, which proves that the proposed filter is a promising choice in the polarization filter field.

## 4. Fabrication and Tolerance Discussion

Besides, the fabrication of plasmonic PCF filters is worth mentioning. There are generally two important steps. First, PCFs are typically fabricated by the conventional stack-and-draw technology [43]. Second, the chemical vapor deposition [44] is adopted, the fabricated fiber end of the neck-down is connected to a syringe, allowing suction of the reaction mixture with metal ions through the unblocked holes. The reaction mixture was maintained at a suitable temperature, then precise control of the rate of metal deposition to achieve the designed deposit thickness, later, the residue must be removed. Recently, Azman et al. chose a different approach for fabricating a broadband copper-filled PCF based polarization filter [45]. In their design, the smaller diameter copper rod was placed into a capillary to make a copper cane, then other capillaries and copper cane were stacked in desired structure for the perform. Finally, drawing the perform to the desired dimension. The proposed filter can also be fabricated by a similar method. Therefore, our PCF filter can be completely fabricated by the modern technology.

At last, the effects on confinement loss and resonance wavelength with the variation of *d*_1_ and *Λ* are considered. Table 3 shows that the effect of ±3% variation of *Λ* on loss and resonance wavelength is more obvious than that of ±3% variation of *d*_1_. Thanks to only one resonance peak in the entire investigated range, the variation in the loss has little effect, and coupled with technological innovation, the wavelength drift could be reduced, so these adverse effects will be tailored precisely.

## 5. Conclusions

In summary, we propose a simple and high-performance plasmonic PCF-filter with dual-ring gold layer by FEM. The effect of the three structural parameters *d*_2_, *d_3,_* and *t* on the performance of the proposed filter is investigated numerically. The simulation results show that when *Λ* is 2 μm, *d*_1_ is 1.2 μm, *d*_2_ is 2.2 μm, *d*_3_ is 1.5 μm, and *t* is 100 nm, the difference in energy between the two polarization directions exceeds 10 times and the confinement loss in the x-pol is less than that in the y-pol direction in the investigated range, which is suitable to realize a broadband polarization filter. When the fiber length is 2 mm, its ER can reach −478 dB and the bandwidth of ER better than −20 dB is 750 nm (1.05–1.8 μm) covering the common communication wavelengths of 1.31 μm and 1.55 μm. It also has low insertion loss of 0.11 dB at 1.31 μm and 0.04 dB at 1.55 μm. Additionally, our filter has a high fabrication feasibility and better tolerance. Therefore, the proposed filter could be a promising choice inn fiber sensors, bio-medical, and fiber communication field in the future.

## Figures and Tables

**Figure 1 micromachines-11-00470-f001:**
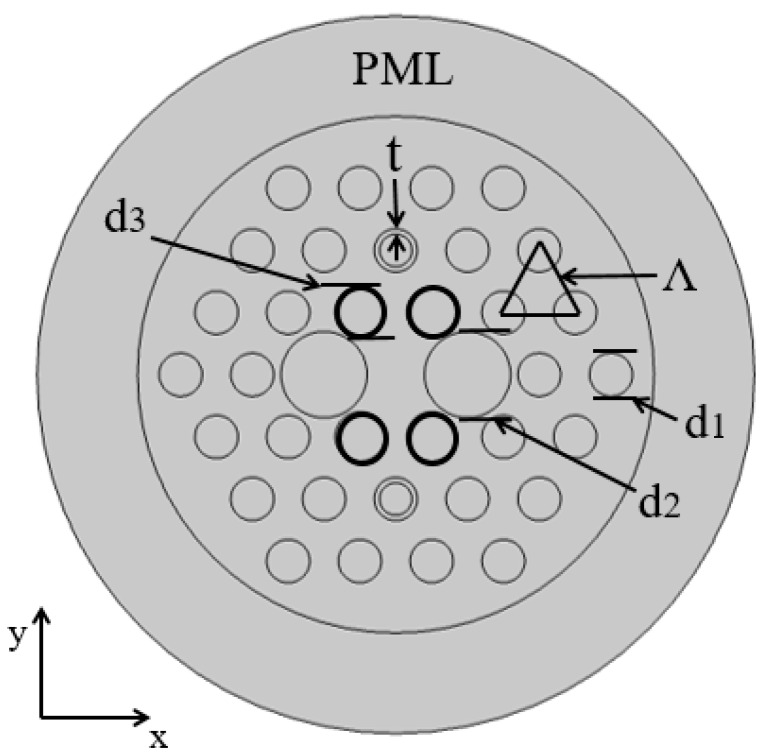
Cross-sectional view of the proposed PCF filter.

**Figure 2 micromachines-11-00470-f002:**
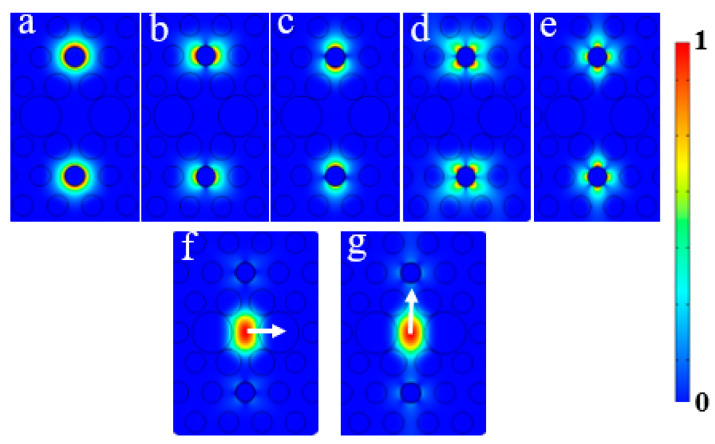
The electric field distributions: (**a**) 0-SPP mode; (**b**) 1-SPP1mode; (**c**) 1-SPP2mode; (**d**) 2-SPP1 mode; (**e**) 2-SPP2 mode; (**f**) x-pol fundamental mode (x-FM); (**g**) y-pol fundamental mode (y-FM).

**Figure 3 micromachines-11-00470-f003:**
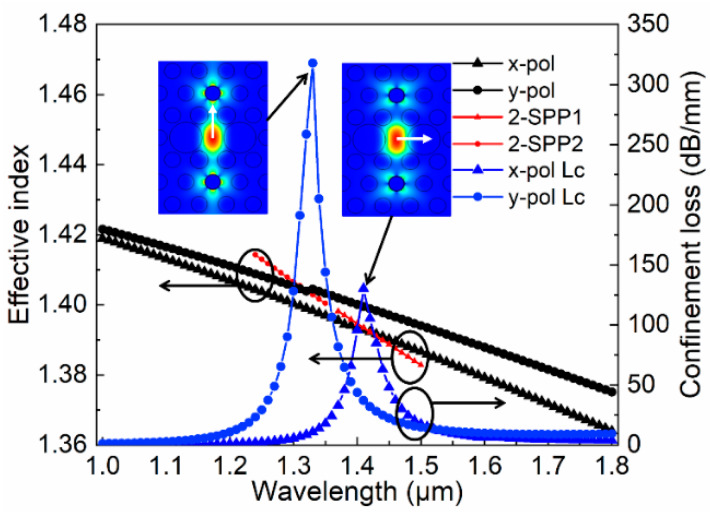
Dispersion relationship between FM and SPP mode for the proposed PCF filter: **left**: y-FM and 2-SPP2 mode; **right**: x-FM and 2-SPP1 mode.

**Figure 4 micromachines-11-00470-f004:**
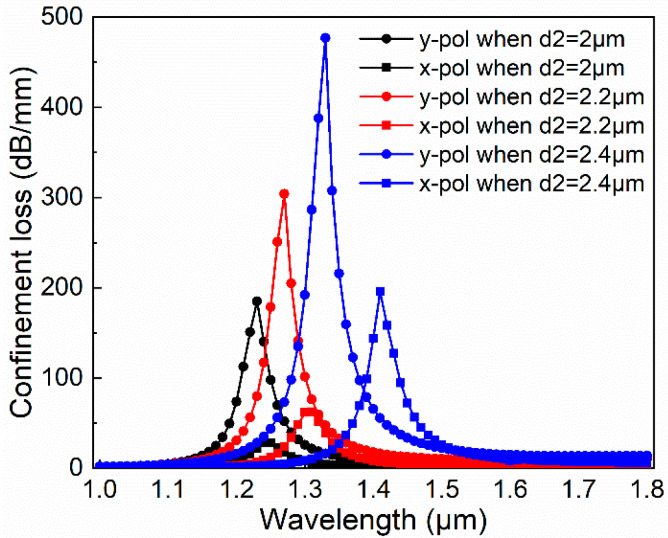
Dependence of transmission spectra in the x- and y-pol direction with different *d*_2_ when *d*_1_ is 1.2 μm *d*_3_ is 1.2 μm, and *t* is 100 nm.

**Figure 5 micromachines-11-00470-f005:**
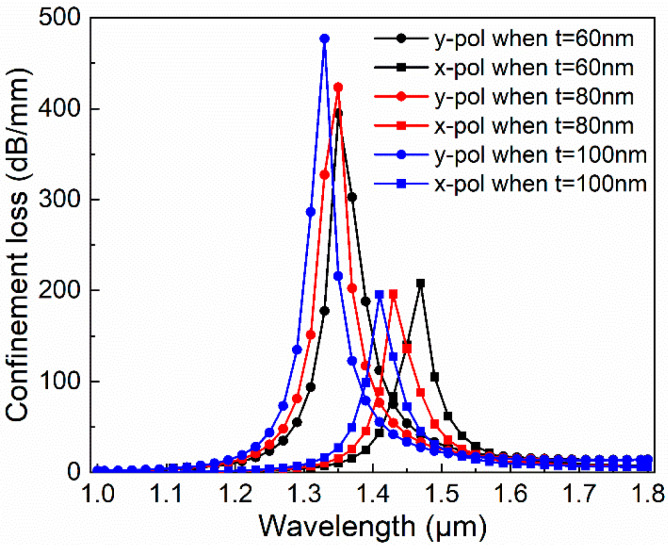
Dependence of transmission spectra in the x- and y-pol direction with different gold film thickness *t* when *d*_1_ is 1.2 μm, *d*_2_ is 2.2 μm, and *d*_3_ is 1.2 μm.

**Figure 6 micromachines-11-00470-f006:**
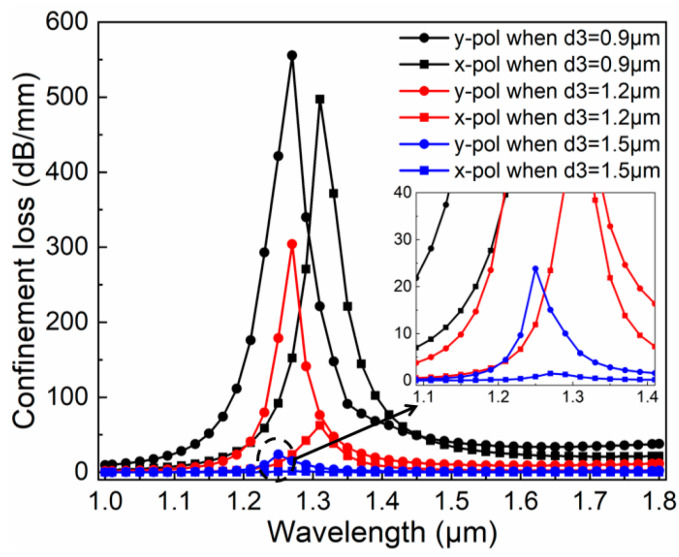
Dependence of transmission spectra in the x- and y-pol direction with different *d*_3_ when *d*_1_ is 1.2 μm, *d*_2_ is 2.2 μm, and *t* is 100 nm.

**Figure 7 micromachines-11-00470-f007:**
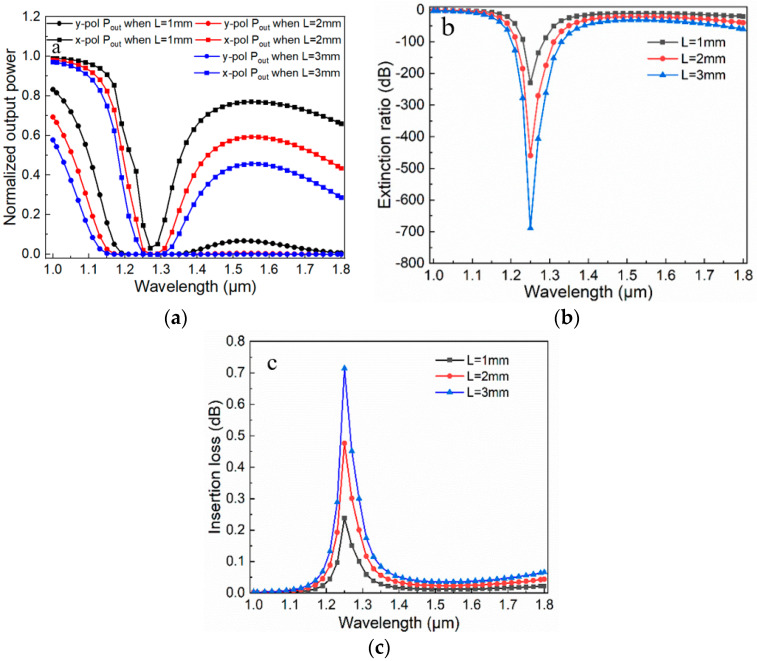
Performance parameters versus wavelength with different fiber lengths *L* for the proposed filter. (**a**) Normalized output power versus wavelength. (**b**) Extinction ratio versus wavelength. (**c**) Insertion loss versus wavelength.

**Figure 8 micromachines-11-00470-f008:**
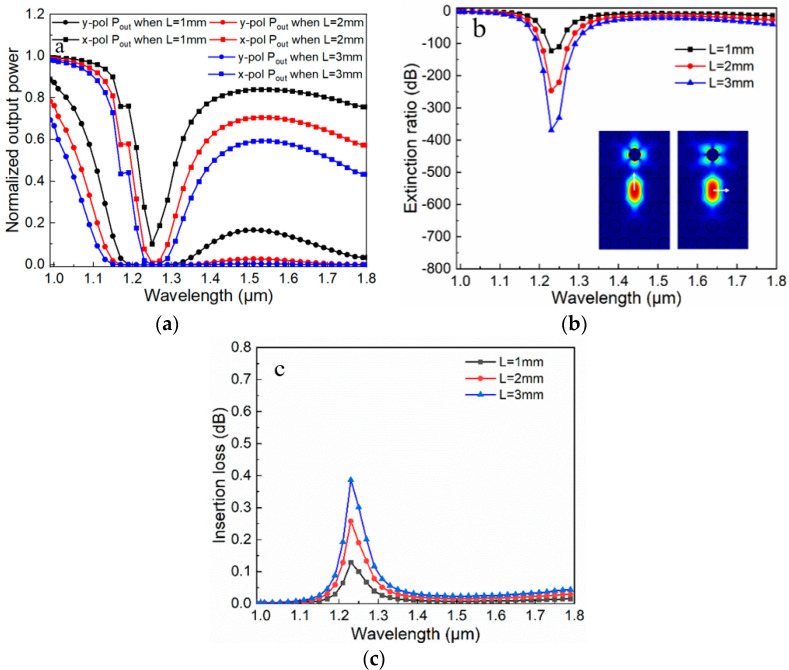
Performance parameters versus wavelength with different fiber length *L* for the single-ring structure filter. (**a**) Normalized output power versus wavelength; (**b**) Extinction ratio versus wavelength, and resonance modes for single-ring structure filter; (**c**) Insertion loss versus wavelength.

**Table 1 micromachines-11-00470-t001:** Detailed parameters in the Drude–Lorentz (DL) model.

$\varepsilon_{\infty }$	Δε	$\omega_{D}/2\pi$ **(THz)**	$\gamma_{D}/2\pi$ **(THz)**	$\Omega_{L}/2\pi$ **(THz)**	${Г}_{L}/2\pi$ **(THz)**
5.9673	1.09	2113.6	15.92	650.07	104.86

**Table 2 micromachines-11-00470-t002:** Performance comparison between the proposed filter and prior filters.

References	Max. Extinction Ratio (dB)	Insertion Loss (dB)	Bandwidth (nm)	Length (mm)
[16]	−160	N/A	500	0.1
[17]	−200.8	0.07(1.55μm)	780	2
[18]	−23.2	N/A	45 (1.3 μm) 100 (1.52 μm)	0.023
[29]	−181	N/A	430	3
[38]	−66.6	N/A	>420	0.3
[39]	~−255	N/A	100	0.2
[40]	231	N/A	224 (1.31 μm) 504 (1.56 μm)	1
[41]	−103	N/A	500	0.8
[42]	~−102	N/A	1010	0.2
This paper	−478	0.11(1.31μm) 0.04(1.55μm)	750	2

**Table 3 micromachines-11-00470-t003:** Fabrication tolerance for the structural parameters of *d*_1_ and *Ʌ*.

Structural Parameters Variation	Loss Variation (α-α_0_)/α	Res. Wav. Variation |λ-λ_0_|(nm)
+3% of *d*_1_	2.3%	2
−3% of *d*_1_	−3.1%	2
+3% of *Λ*	4.3%	5
−3% of *Λ*	−5.2%	4

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
