# Peer review of "Broadband Plasmonic Polarization Filter Based on Photonic Crystal Fiber with Dual-Ring Gold Layer"

_micromachines, 2020, doi:10.3390/mi11050470_

Round 1

Reviewer 1 Report

The paper by N. Chen et al. considers the theoretical assessment to modus operandi of a polarization filter based on a hexagonal photonic crystal fiber with twin central Au-coated pores. As claimed by the authors, due to high plasmonic losses between different light polarization states, the proposed filter design possesses the best performance, low insertion loss and broadband operation range among other similar filters. The paper presented is of moderate scientific soundness, however, the results may have interest to the readers and deserve the publication in MDPI Micromachines.

Several points that may help to improve the paper quality and readability:

  • I strongly recommend to polish the English because it is hard sometimes to follow the authors’ logic;
  • Abstract, line 27: It will be fair referencing the pioneering work by R. Ritchie related the SPPs excitation (R.H. Ritchie, Phys. Rev. 106, 874 (1957));
  • Abstract, line 30: Please specify that SPP refers to the EM-waves propagating along the “metal-dielectric interface” because it demands the plasmon-polariton coupling;
  • Abstract, line 62: The abstract lacks the study motivation by the authors. The novelty and originality of the proposed filter design should be clearly stated here.
  • Section 2, line 81: Along with the Sellmeier equation for RI of fused silica I recommend to give the numerical values of RI in the wavelength range considered;
  • Section 3, line 144: Replace “Fig.5” by “Fig.4”;
  • In Section 3 the authors present the study for filter optimization via varying several structural parameters (d2, d3, t). However in the simulations, these parameters are changed separately and independently from each other. I believe there should be mutual effect between these parameters for filter performance. How it can be accounted for?
  • In fig.4 to fig.6 the graphical insets are desirable showing schematic of the filter with the parameter considered;
  • In Table 2 for Ref.[16] I found the bandwidth parameter should be 780 nm;

Author Response

The paper by N. Chen et al. considers the theoretical assessment to modus operandi of a polarization filter based on a hexagonal photonic crystal fiber with twin central Au-coated pores. As claimed by the authors, due to high plasmonic losses between different light polarization states, the proposed filter design possesses the best performance, low insertion loss and broadband operation range among other similar filters. The paper presented is of moderate scientific soundness, however, the results may have interest to the readers and deserve the publication in MDPI Micromachines.

Several points that may help to improve the paper quality and readability:

Comment 1:  I strongly recommend to polish the English because it is hard sometimes to follow the authors’ logic;

Response: Thank you for approval. We are very sorry for obscure expression in this paper, we have re-read the paper and have corrected the English expression with the help of a native speaker. Revisons in the full paper have been marked in red. Thank you for your suggestions.

Comment 2:  Abstract, line 27: It will be fair referencing the pioneering work by R. Ritchie related the SPPs excitation (R.H. Ritchie, Phys. Rev. 106, 874 (1957));

Response: Thank you for your comments. This reference has been added and cited which have been marked in red on L293.

Comment 3:  Abstract, line 30: Please specify that SPP refers to the EM-waves propagating along the “metal-dielectric interface” because it demands the plasmon-polariton coupling;

Response: Thank you for your guidance. We have corrected the statements that have been marked in red on L33 in Section 1.

Comment 4:  Abstract, line 62: The abstract lacks the study motivation by the authors. The novelty and originality of the proposed filter design should be clearly stated here.

Response: Thank you for your suggestions. We have added relevant text for motivation and novelty of this design on Abstract and L65-L67 in Section 1.

Comment 5:  Section 2, line 81: Along with the Sellmeier equation for RI of fused silica I recommend to give the numerical values of RI in the wavelength range considered;

Response: Thank you for your suggestions. We have added statements about RI of fused silica on L89 in Section 2.

Comment 6:  Section 3, line 144: Replace “Fig.5” by “Fig.4”;

Response: Thank you for your comments. We are sorry for our negligence here. We have corrected it and marked it in red on L152 in Section 3.

Comment 7:  In Section 3 the authors present the study for filter optimization via varying several structural parameters (d2, d3, t). However, in the simulations, these parameters are changed separately and independently from each other. I believe there should be mutual effect between these parameters for filter performance. How it can be accounted for?

Response: Thank you for your comments. The structure of the polarization filter is optimized in order of parameters. By the control variates method, we always get an optimal parameter and fix it, then optimize the next parameter. Your comment is very valuable, but mutual effect between these parameters requires a lot of calculation to verify, we will consider this in our following work. Thank you for your suggestions again.

Comment 8: In fig.4 to fig.6the graphical insets are desirable showing schematic of the filter with the parameter considered;

Response: Thank you for your suggestions. We are sorry we didn’t express clearly, and we have added correct text description in fig.4 to fig.6.

Comment 9: In Table 2 for Ref.[16] I found the bandwidth parameter should be 780 nm;

Response: We are sorry for our negligence here. We have corrected this mistakes on L254 in Section 3.

Reviewer 2 Report

The paper “Broadband Plasmonic Polarization Filter Based on Photonic Crystal Fiber with Dual-Ring Gold Layer” reports on a novel SPP based polarization filter design in a PCF. A systematic study has been numerically performed to understand the effect of the various parameters associated with the design. Study has also been performed to include tolerances for the design to be used in future fabrications. Filter performance is compared with other designs in the literature. The work is impactful and beneficial technologically for telecom. I would like to see the paper published after some minor revisions and suggestions. Some improvements and queries regarding the paper:

  1. A recent paper [Chao Wang et al. "A Broadband Single Polarization Photonic Crystal Fiber Filter Around 1.55 μm Based on Gold-Coated and Pentagonal Structure" Plasmonics (2020): https://doi.org/10.1007/s11468-020-01159-x] reports on a similar work. A comparison with this work could improve the impact of this work.
  2. English must be checked for especially in abstract and figure captions.
  3. In some figures the legends and the plots overlap.
  4. In Fig. 8 (b), please mention the inset mode profiles in the caption.

Author Response

The paper “Broadband Plasmonic Polarization Filter Based on Photonic Crystal Fiber with Dual-Ring Gold Layer” reports on a novel SPP based polarization filter design in a PCF. A systematic study has been numerically performed to understand the effect of the various parameters associated with the design. Study has also been performed to include tolerances for the design to be used in future fabrications. Filter performance is compared with other designs in the literature. The work is impactful and beneficial technologically for telecom. I would like to see the paper published after some minor revisions and suggestions. Some improvements and queries regarding the paper:

Comment 1: A recent paper [Chao Wang et al. "A Broadband Single Polarization Photonic Crystal Fiber Filter Around 1.55 μm Based on Gold-Coated and Pentagonal Structure" Plasmonics (2020): https://doi.org/10.1007/s11468-020-01159-x] reports on a similar work. A comparison with this work could improve the impact of this work.

Response: First of all, thank you for your approval. We have listed the performance parameters of the filter proposed by Wang et al. into Table.1. Thank you for the suggestions.

Comment 2: English must be checked for especially in abstract and figure captions.

Response: Thank you for your guidance. We have re-read abstract and figure captions and have corrected English expression. Revisons in the full paper have been marked in red.

Comment 3: In some figures the legends and the plots overlap.

Response: Thank you for your comments. We are sorry that the figures are not clear. The clear figures have been reattached to the article on L233 and L243.

Comment 4: In Fig. 8 (b), please mention the inset mode profiles in the caption.

Response: Thank you for your guidance. The caption of inset mode profiles has been added to the L248.